# Preclinical Evaluation of a Novel Series of Polyfluorinated Thalidomide Analogs in Drug-Resistant Multiple Myeloma

**DOI:** 10.3390/biom14060725

**Published:** 2024-06-19

**Authors:** Blaire E. Barton, Matthew K. Collins, Cindy H. Chau, Hyoyoung Choo-Wosoba, David J. Venzon, Christian Steinebach, Kathleen M. Garchitorena, Bhruga Shah, Eric L. Sarin, Michael Gütschow, William D. Figg

**Affiliations:** 1Molecular Pharmacology Section, Genitourinary Malignancies Branch, Center for Cancer Research, National Institutes of Health, Bethesda, MD 20892, USA; 2Biostatics and Data Management Section, Center for Cancer Research, National Cancer Institute, National Institutes of Health, Bethesda, MD 20892, USA; 3Pharmaceutical Institute, Pharmaceutical & Medicinal Chemistry, University of Bonn, 53121 Bonn, Germany; 4Inova Heart and Vascular Institute, Inova Health System, Falls Church, VA 22042, USA

**Keywords:** multiple myeloma, resistance, immunomodulatory drugs, cereblon, angiogenesis

## Abstract

Immunomodulatory imide drugs (IMiDs) play a crucial role in the treatment landscape across various stages of multiple myeloma. Despite their evident efficacy, some patients may exhibit primary resistance to IMiD therapy, and acquired resistance commonly arises over time leading to inevitable relapse. It is critical to develop novel therapeutic options to add to the treatment arsenal to overcome IMiD resistance. We designed, synthesized, and screened a new class of polyfluorinated thalidomide analogs and investigated their anti-cancer, anti-angiogenic, and anti-inflammatory activity using in vitro and ex vivo biological assays. We identified four lead compounds that exhibit potent anti-myeloma, anti-angiogenic, anti-inflammatory properties using three-dimensional tumor spheroid models, in vitro tube formation, and ex vivo human saphenous vein angiogenesis assays, as well as the THP-1 inflammatory assay. Western blot analyses investigating the expression of proteins downstream of cereblon (CRBN) reveal that Gu1215, our primary lead candidate, exerts its activity through a CRBN-independent mechanism. Our findings demonstrate that the lead compound Gu1215 is a promising candidate for further preclinical development to overcome intrinsic and acquired IMiD resistance in multiple myeloma.

## 1. Introduction

Multiple myeloma (MM) is a hematological malignancy that makes up nearly 2% of all cancers and accounts for more than 12,500 deaths annually [1]. With that said, introducing novel therapeutics, including immunomodulatory drugs (IMiDs) and proteasome inhibitors, has markedly increased patients’ survival. For instance, since the FDA approval of thalidomide in 2006, 5-year overall survival rates have risen from 47% to nearly 60% [2]. Multiple myeloma is characterized by the accumulation of monoclonal plasma cells in the bone marrow, which overproduce a monoclonal immunoglobulin known as M-protein. The disease results in complications such as hypercalcemia, renal insufficiency, anemia, bone deterioration, and immunosuppression [3,4]. Current standard therapies include a variety of treatment classes, with proteasome inhibitors, CD38-targeting antibodies, and IMiDs representing the current standard of care [3,5]. The IMiDs lenalidomide, a second-generation thalidomide analog, and pomalidomide, a third-generation analog of thalidomide, are among these approved agents [3,5]. However, most patients with MM relapse multiple times, with increasingly shorter remissions, and eventually develop disease refractory to all available therapies [3,6]. To improve clinical outcomes for these patients with relapsed-refractory multiple myeloma (RRMM), novel treatment modalities are required to achieve deeper and more durable responses as well as provide long-term disease control.

Angiogenesis, the process of forming new blood vessels from existing vessels, plays a crucial role in many hematological malignancies, including MM [7,8]. The monoclonal plasma cells within the MM bone marrow microenvironment secrete pro-angiogenic factors, and the subsequent increase in vasculature is correlated with increased tumor growth, more frequent relapse, and the development of drug resistance [9]. Additionally, the inflammatory response and increased production of specific pro-inflammatory cytokines have been implicated in the pathogenesis of MM and other hematological malignancies [10]. These findings underscore the need for antiangiogenic and anti-inflammatory therapies within the MM treatment landscape. The anti-angiogenic and anti-inflammatory properties of thalidomide and its two FDA-approved analogs, lenalidomide and pomalidomide, are well-established [11,12,13]. While the two analogs demonstrate increased potency and decreased toxicity when compared to thalidomide, patients still suffer from related side effects, and many develop resistance to these drugs [14,15].

The primary target of thalidomide and its analogs is cereblon (CRBN), the substrate recognition element of the E3 cullin 4-RING ubiquitin ligase complex [16]. The interaction of IMiDs with CRBN triggers the ubiquitination and subsequent proteasomal degradation of the neo-substrates Ikaros (IKZF1) and Aiolos (IKZF3) [17,18], leading to the down-regulation of transcription factors interferon regulatory 4 (IRF4) and c-MYC. Since these transcription factors play a role in promoting the survival and proliferation of the monoclonal MM plasma cells, this IMiD–CRBN interaction explains, in part, the anti-myeloma effects of these compounds [4]. Down-regulation of CRBN has been postulated to be one of the mechanisms of acquired resistance to IMiDs [19]. Genetic alterations in CRBN may also contribute to acquired IMiD resistance, as genomic aberrations have increased incrementally as MM patients progress from newly-diagnosed to lenalidomide-resistant and finally to pomalidomide-resistant [20]. Additionally, since various substrates compete for CRBN binding, the increased expression of alternative CRBN substrates may outcompete IKZF1 and IKZF3, thus reducing their degradation by CRBN and promoting IMiD resistance [21,22]. These mechanisms of IMiD resistance, amongst others not included here, illustrate the need for novel therapeutics for IMiD-refractory MM.

Our laboratories have a long history of investigating the in vitro and in vivo therapeutic efficacies of thalidomide analogs in manty anti-cancer assays [23,24,25,26]. We previously reported on the synthesis and structural characterizations of a novel class of polyfluorinated thalidomide analogs [27]. The current study screened these novel 26 compounds using in vitro, ex vivo, and three-dimensional spheroid assays in commonly used MM models to determine their anti-angiogenic, anti-inflammatory, and anti-cancer properties, particularly in IMiD-refractory MM cell lines. 

## 2. Materials and Methods

### 2.1. Thalidomide Analogs

Analogs were synthesized to a chemical purity of >99.5% (as assessed by LC/MS, ^1^H NMR, and ^13^C NMR) at the University of Bonn, Germany [27]. U.S. patents of these analogs have been filed (U.S. Patent Application No. 63/518,823, filed 10 August 2023).

### 2.2. Cell Lines and Reagents

Human umbilical vein endothelial cells (HUVEC; RRID:CVCL_2959) were purchased from Lonza (Walkersville, MD, USA), cultured in EGM-plus media (Lonza), seeded to collagen-I-coated flasks (Corning Life Sciences, Tewksbury, MA, USA), and used before passage 12. The human leukemic monocytic cell line THP-1 (RRID:CVCL_0006) and human myeloma cell lines RPMI-8226 (RRID:CVCL_0014) and MM1.S (RRID:CVCL_8792) were purchased from American Type Culture Collection (ATCC, Manassas, VA, USA) and grown in RPMI-1640 (Life Technologies, Carlsbad, CA, USA) with 10% FBS and 1% P/S. The human myeloma cell line JJN3 (RRID:CVCL_2078) was purchased from DSMZ (Leibniz Institute DSMZ–German Collection of Microorganisms and Cell Cultures, Braunschweig, Germany) and grown in 40% DMEM and 40% IMDM (Life Technologies, Carlsbad, CA, USA) with 20% FBS. The human myeloma cell line MOLP-8 (RRID:CVCL_2124) was also purchased from DSMZ and grown in RPMI-1640 with 15% FBS and 1% P/S. MM1.S (RRID:CVCL_8792) and U266 (RRID:CVCL_0566) and their respective drug-resistant counterparts MM1/R10R and U266/R10R were kindly provided by Dr. Robert Orlowski (The University of Texas, MD Anderson Cancer Center, Houston, TX, USA) [28]. The MM1.S and MM1/R10R cell lines were grown in RPMI-1640 with 10% FBS and 1% P/S, and the U266 and U266/R10R cell lines were grown in RPMI-1640 with 15% FBS and 1% P/S. All cell lines were cultured in 5% CO_2_ and 95% air at 37 °C and underwent routine cell line authentication and mycoplasma testing through ATCC. Dimethyl sulfoxide (DMSO) was purchased from Sigma Aldrich (St. Louis, MO, USA).

### 2.3. Cell Proliferation Assay

Compounds were tested for their ability to inhibit the proliferation of various cell lines in vitro using a CCK-8 assay. RPMI-8226 (10,000 cells/well), JJN3 (20,000 cells/well), MOLP-8 (20,000 cells/well), MM1.S (R10R and WT) (20,000 cells/well), and U266 (R10R and WT) (20,000 cells/well) cells were plated in 96-well plates and allowed to incubate overnight at 37 °C in 5% CO_2_. An equal amount of treatment media containing the vehicle (0.5% DMSO) or test compounds at a range of concentrations was then plated on top of the seeded cells for 24, 48, or 72 h of incubation. Cell proliferation was evaluated by adding 10 μL of CCK-8 reagent (Dojindo Laboratories, Rockville, MD, USA), incubating for 2–4 h, and measuring the absorbance at 450 nm on a microplate reader (SpectraMax iD3, Molecular Devices, San Jose, CA, USA). 

### 2.4. Three-Dimensional Tumor Spheroid Assay

Three-dimensional (3D) spheroids were cultured as previously described [29]. Briefly, MOLP-8 and RPMI-8226 cells were seeded at 1250 and 800 cells/well, respectively, in 50 µL volumes in 96-well ultra-low attachment plates and allowed to incubate for 30 min at 37 °C in 5% CO_2_, 95% air. The seeded cells were then treated with equal volumes of media containing vehicle or test compounds at a range of concentrations. Following 72 h of incubation, cell viability within the spheroids was evaluated using a CellTiter-Glo 3D Cell Viability Assay (Promega, Madison, WI, USA). An amount of 100 µL of room-temperature CellTiter-Glo 3D reagent was added to each well, and plates were placed on a shaker for 5 min and then allowed to sit at room temperature for 25 additional minutes. Subsequently, luminescence was measured on a microplate reader (SpectraMax iD3) with an integration time of 1000 ms. 

### 2.5. THP-1 Inflammatory Assay

THP-1 macrophages, when stimulated with lipopolysaccharide (LPS) bacteria, secrete the inflammatory cytokine tumor necrosis factor-alpha (TNF-α) [30]. THP-1 cells were plated in 24-well plates at 45,000 cells/well and allowed to attach overnight at 37 °C in 5% CO_2_. THP-1 cells were then pretreated for 1 h with either vehicle (0.5% DMSO) or test compound and subsequently challenged with LPS (1 ng/mL). After three hours, the cells were spun at 12,000 rpm for 7 min and the supernatant was collected. The levels of TNF-α were measured by ELISA using a human TNF-α ELISA kit (cat. no. KHC3011; Thermo Fisher, Waltham, MA, USA) according to manufacturer’s instructions.

### 2.6. Endothelial Cell Tube Formation Assay (Lattice Assay)

An in vitro angiogenesis assay kit was purchased from EMD Millipore (Darmstadt, Germany). An amount of 50 μL/well of ECMatrix was plated onto a 96-well plate and left to set for at least 30 min. To harvest cells, HUVECs were detached using a TryplE Express (Thermo Fisher), spun at 1300 rpm, and resuspended in EGM-plus media. Cells were plated atop the gel (35,000 cells/well) and treated with the vehicle control (0.5% DMSO), the positive control (30 μM CPS49), or test compounds in a range of concentrations. The wells were imaged after 18 h of incubation. Tubule formation was quantified using ImageJ (Version 1.54j). 

### 2.7. Human Saphenous Vein (HSV) Angiogenesis Assay

The anti-angiogenic effects of the test compounds were evaluated in the human saphenous vein angiogenesis model. Specimens of human saphenous veins were harvested and collected during surgery on a study protocol approved by the Institutional Review Board at the INOVA Heart and Vascular Institute (WCG IRB Tracking Number: 20210290; Protocol Number: U19-01-3349; Fairfax, VA, USA). Subsequently, 24-well tissue culture plates were covered with 350 μL of Matrigel (Corning Life Sciences) and allowed to set for one hour at 37 °C. Following the excision of fibroadipose tissue, the veins were cut into 2 mm cross sections, placed on the Matrigel-coated wells, and layered with additional Matrigel (450 μL). These were then allowed to set for one hour, after which the cross-sectional rings were covered with EGM-II endothelial cell growth media (Lonza) and incubated under 5% CO_2_/95% air at 37 °C overnight. EGM-II consists of endothelial cell basal medium-2 (EBM-II) and endothelial cell growth medium-2 SingleQuots supplements and growth factors. The next day, the rings were treated with media containing the vehicle control (0.5% DMSO), a known angiogenesis inhibitor (TNP-470) as the positive control, or test compounds in a range of concentrations. The rings were incubated for a total of 14 days, being re-treated every 2 to 3 days, with all growth factors removed from the medium starting on day 7. The rings were then imaged on day 15 using an EVOS scope. This was independently replicated two times using specimens from 2–3 different donors. The area of angiogenic sprouting, reported in square pixels, was quantified using Adobe Photoshop. Data are presented as percent growth based on the vehicle control, which was normalized to 100% growth.

### 2.8. Western Blot Analysis

MM1.S and MOLP-8 cells were plated in 6-well plates at 1.5 × 10^6^ cells/well and allowed to incubate overnight at 37 °C in 5% CO_2_, 95% air. The cells were treated with the vehicle with or without drugs for 24 h. Cells were collected from the 6-well plates, washed with ice-cold dPBS, centrifuged at 1300 rpm, then lysed with RIPA lysis buffer (Sigma Aldrich) and complete protease inhibitor (Catalog no. 25955-11, Nacalai USA, San Diego, CA, USA). Samples were left on ice for 10 min, briefly vortexed, and returned to the ice for another 10 min. The lysed cells were then centrifuged at 13,000 rpm at 4 °C for 10 min. The protein-containing supernatant was carefully removed, and the amount of protein in the supernatant was quantified using a Pierce bicinchoninic acid (BCA) assay (Thermo Fisher Scientific, Waltham, MA, USA). Samples were run on 4–20% SDS-PAGE Mini-PROTEAN TGX^TM^ Precast Gels (Bio-Rad Laboratories Inc., Hercules, CA, USA) and separated by electrophoresis at 80 V for 90 min. Gels were transferred to 0.2 μm nitrocellulose membranes using the Mini Trans-Blot Turbo semi-dry transfer system (Bio-Rad). Nitrocellulose membranes were blocked with 5% nonfat dry milk in 0.1% TBST for 60 min and probed overnight at 4 °C with antibodies against Aiolos (CST catalog no. 15103; 1:1000; Cell Signaling Technology, Danvers, MA, USA), Ikaros (CST catalog no. 5443; 1:100; Cell Signaling Technology, Danvers, MA, USA), and GAPDH (SC-47724; 1:2000; Santa Cruz Biotechnology, Dallas, TX, USA). Membranes were washed 3 times with 0.1% TBST and incubated in Goat Anti-Mouse IRDye 680RD (1:10,000, LI-COR 925-68070; LI-COR, Lincoln, NE, USA) and Goat Anti-Rabbit IRDye 800CW (1:10,000, Licor 925-32211) for one hour at room temperature. Blots were imaged on an LI-COR Odyssey Fc (LI-COR).

### 2.9. Statistics

All results are presented as mean ± SEM. For the TNF-α ELISA and the human saphenous vein angiogenesis model, treatment groups were compared using the Mann–Whitney *U* test, followed by the Hochberg method for adjusting p-values for multiple comparisons. All other data were compared utilizing a one-way analysis of variance (ANOVA) with different variances of residuals across concentrations to account for a possibility of a heterogeneity of variance across concentrations and employing the Satterthwaite method to adjust the degrees of freedom of each pairwise difference to account for the varying standard deviations across concentrations. The data were representative of at least three independent experiments with at least three replicates per experiment. In all the figures, any cases in which the concentrations are grouped in a bracket indicate that all of the values within the bracket are individually statistically different from the vehicle control.

## 3. Results

### 3.1. Effects of Thalidomide Analogs on In Vitro Proliferation of Intrinsically IMiD-Resistant MM Cells

We first screened the library of 26 thalidomide analogs to assess for anti-cancer activity in commonly used MM models. The two MM cell lines, RPMI-8226 and JJN3, have been shown to carry intrinsic resistance to lenalidomide [31]. Prior to dose-response drug treatment experiments, we first conducted a preliminary screening of the 26 compounds at the single 10 μM concentration for their ability to inhibit the growth of the two intrinsically IMiD-resistant MM cell lines and to identify effective compounds. In the presence or absence of analogs, RPMI-8226 and JJN3 cells were incubated for 72 and 24 h, respectively, to allow for at least one doubling time to pass for each respective cell line, after which we conducted a CCK-8 cell proliferation assay to determine cytotoxicity. Four compounds (Gu1210, Gu1213, Gu1214, and Gu1215; Figure 1) exhibited the most potent activity in both RPMI-8226 and JJN3 cells, with each analog inhibiting growth by ≥90% and ≥50%, respectively (Figure 2A,B). 

We conducted further testing on these four lead analogs, determining the effects of each compound on cell proliferation at a range of concentrations (0.10 to 20 μM) in both RPMI-8226 and JJN3 cells. All four analogs inhibited cell proliferation dose-dependently in both IMiD-resistant cell lines (Figure 2C,D). In RPMI-8226 cells, Gu1210 and Gu1214 demonstrated IC_50_ values of 2.5 µM and 3.0 µM, respectively, while Gu1213 and Gu1215 more potently inhibited the growth of cells, with an IC_50_ of 2.4 µM and 1.6 µM, respectively (Figure 2C). In JJN3 cells, Gu1215 was the most potent of the four analogs, inhibiting cell proliferation with a demonstrated IC_50_ value of 3.4 µM [vs. Gu1213 (5.3 µM), Gu1210, and Gu1214 (IC_50_ values > 10 µM)] (Figure 2D). The four lead analogs were also tested in the MOLP-8 multiple myeloma cell line, which carries intrinsic resistance to lenalidomide as well as some inherent resistance to pomalidomide [32]. In this cell line, compounds were tested at 0.1 µM, 1 µM, 5 µM, and 10 µM. As in the other intrinsically IMiD-resistant cells lines, each of the analogs inhibited cell proliferation in a dose-dependent manner. Gu1214 demonstrated the most potent inhibition of cell proliferation in this cell line, exhibiting an IC_50_ of 1.00 µM, followed by Gu1210, Gu1215, and Gu 1213, with IC_50_s of 1.16, 1.23, and 1.91, respectively (Figure 2E).

### 3.2. Effects of Thalidomide Analogs on In Vitro Proliferation of MM Cells with Acquired IMiD Resistance

We investigated the effects of the four lead compounds (Gu1210, Gu1213, Gu1214, and Gu1215) in acquired IMiD-resistant MM cell lines. These lines were developed by the continuous exposure of cells to increasing concentrations of lenalidomide. We used two paired cell lines: wild type (WT) MM1.S and WT U266 and their lenalidomide-resistant counterparts MM1.R10R and U266/R10R, respectively. The cell lines were tested with the lead compounds at a concentration range of 50–10 µM. The four lead analogs each inhibited cell proliferation in a dose-dependent manner in all cell lines (Figure 3A–D). In MM1/R10R cells, Gu1210, Gu1213, and Gu1214 all showed similar IC_50_ values of 5.6 µM, 4.9 µM, and 5.0 µM, respectively, while Gu1215 inhibited proliferation most potently, with an IC_50_ of 2.7 µM (Figure 3A). In the parental MM1.S cell line, Gu1210 produced the most significant growth inhibition, yielding an IC_50_ value of 0.45 µM (Figure 3B). The other three analogs were similarly potent, with IC_50_ values of 0.60 µM for Gu1213, 0.52 µM for Gu1214, and 0.78 µM for Gu1215 (Figure 3B). In U266/R10R cells, Gu1213, Gu1214, and Gu1215 inhibited proliferation with respective IC_50_ values of 2.67 µM, 3.10 µM, and 3.09 µM, while Gu1210, the most potent lead analog in this cell line, demonstrated inhibition with an IC_50_ value of 1.71 µM (Figure 3C). In WT U266, Gu1215 exhibited the most potent inhibition of cell proliferation, with an IC_50_ of 2.23 µM (Figure 3D). Gu1210, Gu1213, and Gu1214 each exhibited slightly less potent inhibitions, with IC_50_ values of 2.34 µM, 3.33 µM, and 3.50 µM, respectively (Figure 3D).

### 3.3. Effects of Thalidomide Analogs on 3D Myeloma Spheroid Growth

Following the evaluation of the polyfluorinated thalidomide analogs in cell proliferation assays, we analyzed their anti-cancer activity in three-dimensional spheroid models in the two IMiD-resistant MM cell lines, MOLP-8 and RPMI-8226. Following the 72-h treatment of the MOLP-8 spheroids with the four leads at varying concentrations [0.1–10 µM], cell viability within the spheroids was reduced in a dose-dependent manner. Gu1213 produced the most potent inhibition of spheroid growth, with an IC_50_ of 0.85 µM, followed closely by Gu1210 (0.89 µM) > Gu1214 (1.06 µM) > Gu1215 (1.12 µM) (Figure 4A). After the 72-h treatment of the RPMI-8226 spheroids, Gu1215 produced the most pronounced inhibition of spheroid growth, with an IC_50_ of 0.99 µM. Gu1210, Gu1213, and Gu1214 exhibited IC_50_ values of 1.73 µM, 1.97 µM, and 2.30 µM, respectively, in this spheroid model (Figure 4B). 

### 3.4. In Vitro Inflammatory Response to Thalidomide Analogs

We then investigated the compounds’ abilities to inhibit the expression of the inflammatory cytokine TNF-α from THP-1 cells, induced by challenging them with lipopolysaccharides (LPS) from *E. coli*. Thalidomide was included as a comparator. After pretreating with the vehicle (0.5% DMSO) or compound for 1 h, THP-1 cells were challenged with LPS for 3 h, and the extent of TNF-α expression was determined by ELISA. For this assay, we set screening concentrations of 5 µM and 10 µM based on initial assays conducted to assess the potency of these compounds. Out of all 26 compounds tested at the screening concentrations, four analogs (Gu1210, Gu1213, Gu1214, and Gu1215) demonstrated increased potencies and were selected for further testing. Each compound was tested at concentrations of 250 nM, 500 nM, and 750 nM, and all four analogs inhibited TNF-α expression in a dose-dependent fashion. Of the four compounds tested at these doses, Gu1215 exhibited the greatest potency in this assay (IC_50_ = 161 nM). Gu1210, Gu1213, and Gu1214 demonstrated IC_50_ values of 338 nM, 301 nM, and 253 nM, respectively (Figure 5).

### 3.5. Effects of Thalidomide Analogs on Endothelial Tube Formation

We assessed the compounds’ anti-angiogenic properties with an endothelial tube formation (lattice) assay, a frequently used in vitro angiogenesis assay [33]. CPS49, a thalidomide analog with potent anti-angiogenic activity [24,34], served as a positive control, while thalidomide was included as a comparator. The ability of the human umbilical vein endothelial cells (HUVECs) to form tubules was not markedly inhibited by thalidomide at 100 μM (Figure 6A), consistent with previously published data [35]. Also consistent with previous data [24,34], CPS49 at 30 μM significantly reduced tube formation by >80% (Figure 6A). We treated HUVECs with the four lead compounds (1–10 µM) to determine the effect of these compounds in this assay. All four analogs demonstrated >50% inhibition of tube formation at 5 µM (Gu1214 ≈ Gu1215 ≈ Gu1213 > Gu1210) (Figure 6A). Representative images of tube formation are shown in Figure 6B.

### 3.6. Anti-Angiogenic Effects of Thalidomide Analogs in the Ex Vivo Human Saphenous Vein Model

We further assessed the anti-angiogenic properties of the four lead compounds (Gu1210, Gu1213, Gu1214, and Gu1215) in an ex vivo human saphenous vein model of angiogenesis. TNP-470, a known angiogenesis inhibitor, served as a positive control, while thalidomide was included as a comparator. The analogs were tested at 10 µM and 25 µM and dosed repeatedly over a two-week period. On day 15, we assessed the extent of microvessel outgrowth in each test group. At 50 µM, TNP-470 inhibited microvessel outgrowth by 99% (Figure 7A). Outgrowth was inhibited by 20% by thalidomide at 100 µM (Figure 7A). All four lead compounds demonstrated dose-dependent inhibition of microvessel outgrowth with Gu1213 and Gu1215 exhibiting the most potent activity (Figure 7A). At 10 µM, Gu1215 produced the most potent inhibition, reducing microvessel outgrowth by 83%. Gu1213 was the next most potent analog, inhibiting outgrowth by 80%. At 25 µM, Gu1213 and Gu1215 reduced microvessel outgrowth by >98% (Figure 7A). Representative images of microvessel outgrowth are shown in Figure 7B.

### 3.7. Effects of Treatment with Lead Thalidomide Analog on the Expression of Proteins Downstream of CRBN

Next, we assessed the effect of the primary lead compound (Gu1215) on proteins downstream of CRBN, the known molecular target of lenalidomide and pomalidomide [16]. IMiD-induced conformational changes in CRBN promote binding and subsequent degradation of two protein substrates, Ikaros (IKZF1) and Aiolos (IKZF3). After a 24-h treatment of MM1.S cells with lenalidomide or pomalidomide, both IKZF1 and IKZF3 exhibited decreased expressions in the Western blot analysis (Appendix A), consistent with results from previous reports. In contrast, neither IKZF1 nor IKZF3 demonstrated decreased expressions following a 24-h treatment with Gu1215 at concentrations of 0.1 µM, 0.5 µM, or 1 µM (Appendix A). Likewise, a 24-h treatment of MOLP-8 cells with lenalidomide or pomalidomide reduced IKZF1/3 expression, but the treatment with Gu1215 did not affect the expression of either substrate at 0.1, 0.5, or 1 µM (Appendix A). 

## 4. Discussion

We screened a novel class of polyfluorinated thalidomide analogs to determine their anti-angiogenic, anti-inflammatory, and anti-cancer activity in lenalidomide- and pomalidomide-refractory MM cells. We identified four lead thalidomide analogs (Gu1210, Gu1213, Gu1214, and Gu1215) exhibiting varying degrees of anti-inflammatory, anti-angiogenic, and anti-cancer properties. 

We assessed the ability of each lead compound to inhibit the proliferation of several MM cell lines, particularly those that carry intrinsic or acquired resistance to IMiDs (e.g., lenalidomide and pomalidomide). We found that all compounds reduced the extent of cell proliferation in each cell line tested, with Gu1215 as the lead candidate in inhibiting cell proliferation in most MM cell lines. In our three-dimensional spheroid models, all compounds reduced tumor spheroid growth in a dose-dependent manner, with Gu1215 exhibiting potent activity in RPMI-8226 spheroids and Gu1213 exhibiting the greatest potency in the MOLP-8 spheroid model.

We also investigated the ability of the lead compounds to inhibit the expression of TNF-α using a TNF-α ELISA, with Gu1215 exhibiting the most potent anti-inflammatory effects. The anti-inflammatory properties of the lead compounds tested here compared favorably to structurally similar thalidomide analogs; each compound tested demonstrated more significant inhibition of TNF-α expression at lower concentrations (500 nM) than those tested previously at 25 µM [36].

We previously reported on the anti-angiogenic effects of these 26 analogs in the ex vivo rat aorta ring (RAR) model of angiogenesis and identified Gu1210, Gu1213, Gu1214, and Gu1215 as possessing anti-angiogenic activity [27]. In the current study, each of our four lead analogs performed similarly in the well-established endothelial cell tube formation assay compared to structurally related analogs previously tested at 5 µM [23]. We further evaluated these analogs in a human saphenous vein assay and confirmed that Gu1210, Gu1213, Gu1214, and Gu1215 are potent angiogenesis inhibitors, consistent with our previous RAR data [27]. Gu1215 and Gu1213 produced the most marked inhibition, reducing microvessel outgrowth by over 80% at 10 µM (Figure 7A). 

The precise mechanisms of action by which thalidomide and its analogs exert anti-angiogenic effects are not entirely understood. Initial studies identified cereblon (CRBN) as the protein target of thalidomide and its close analogs lenalidomide and pomalidomide [16,37,38]. However, we recently showed that the loss of CRBN alone does not eliminate thalidomide’s anti-angiogenic activity [39]. In subsequent studies using, for example, in silico CRBN-docking experiments, we found that enhanced cereblon binding does not correlate with anti-angiogenic effects of thalidomide analogs [40,41]. A recent study using FRET pairing to investigate the CRBN binding abilities of a variety of phthalimides, *N*-substituted with 5- and 6-membered rings, found that at least one imide carbonyl group is required for binding [42]. Gu1215 (Figure 1), investigated in this study, also bears a cyclic substituent, which however lacks a carbonyl group or any moiety to be engaged in polar or hydrogen–bond interactions. Nevertheless, Gu1215 demonstrated potent anti-angiogenic properties in both angiogenesis assays. Taken together, these data suggest that IMiD-induced antiangiogenic effects may be exerted through a non-CRBN protein target. It has been previously established that acquired resistance to lenalidomide is associated with a significant decrease in or absence of CRBN protein expression compared to isogenic lenalidomide-sensitive cell lines [32,43,44]. Conversely, baseline CRBN expression levels are not correlated with IMiD sensitivity, as intrinsically IMiD-resistant MM cell lines have baseline CRBN expression levels comparable to their IMiD-sensitive counterparts [32]. Additionally, the treatment of MM1.S and MOLP-8 cells with Gu1215 did not affect IKZF1 and IKZF3, the downstream proteins of CRBN that exhibited decreased expression when treated with lenalidomide and pomalidomide (Figure 3E). Therefore, our data suggest that our lead compounds may have been exerting their anti-myeloma effects through a non-CRBN target and a mechanism that is yet to be understood. Further genome-wide CRISPR knockout studies are needed to identify the protein target of our novel thalidomide analogs. Given its efficacy in all assays conducted in this study, particularly against IMiD-resistant MM cell lines, Gu1215 also warrants further investigation as a potential drug candidate for lenalidomide- and pomalidomide-resistant multiple myeloma.

## 5. Patents

U.S. patents of these analogs have been filed (U.S. Patent Application No. 63/518,823 filed 10 August 2023).

## Figures and Tables

**Figure 1 biomolecules-14-00725-f001:**
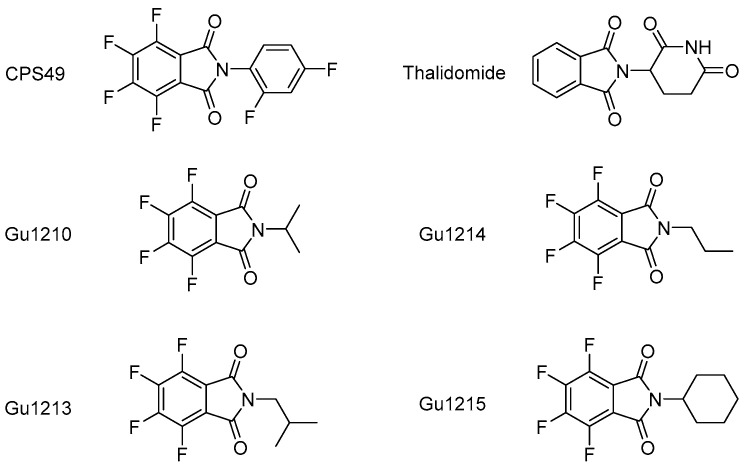
Structures of thalidomide, the first-generation thalidomide analog CPS49, and the thalidomide analogs Gu1210, Gu1213, Gu1214, and Gu1215.

**Figure 2 biomolecules-14-00725-f002:**
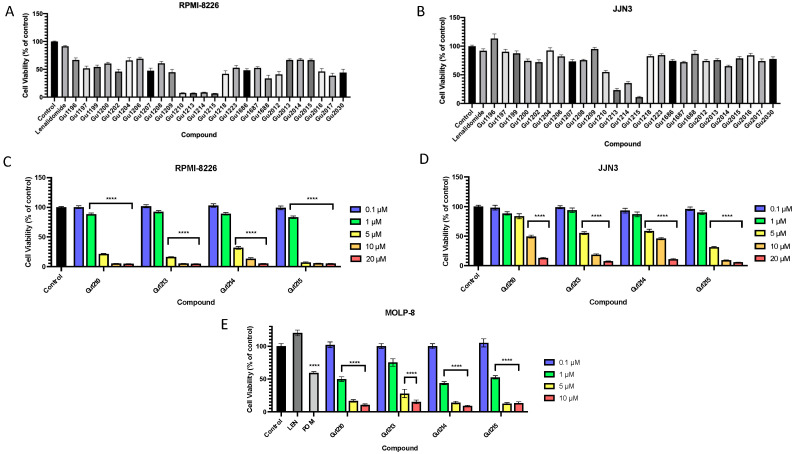
Treatment with Gu compounds inhibited multiple myeloma cell proliferation in vitro. (**A**) The RPMI-8226 cell proliferation during 72-h treatment was most potently inhibited by Gu1210, Gu1213, Gu1214, and Gu1215 at a concentration of 10 μM compared to that of the vehicle control (0.5% DMSO). (**B**) The JJN3 cell proliferation during 24-h treatment was most potently inhibited by Gu1210, Gu1213, Gu1214, and Gu1215 at a concentration of 10 μM compared to that of the vehicle control (0.5% DMSO). (**C**) The 72-h treatment of RPMI-8226 cells with four lead analogs at a range of concentrations (0.1 μM–20 μM) showed a dose-dependent response, with Gu1215 exhibiting the most potent inhibition (**** *p* < 0.0001). (**D**) The 24-h treatment of JJN3 cells with the four lead analogs at a range of concentrations (0.1 μM–20 μM) showed a dose-dependent response, with Gu1215 exhibiting the most potent inhibition (**** *p* < 0.0001). Note: a 5 µM dose represents two independent replicates. (**E**) The 72-h treatment of MOLP-8 cells with the four lead analogs at a range of concentrations (0.1 μM–10 μM) showed dose-dependent responses, with Gu1214 exhibiting the most potent inhibition (**** *p* < 0.0001). These data are representative of at least three independent experiments with at least three replicates per experiment. In all the figures, any cases in which the concentrations are grouped in a bracket indicate that all of the values within the bracket are individually statistically different from the vehicle control.

**Figure 3 biomolecules-14-00725-f003:**
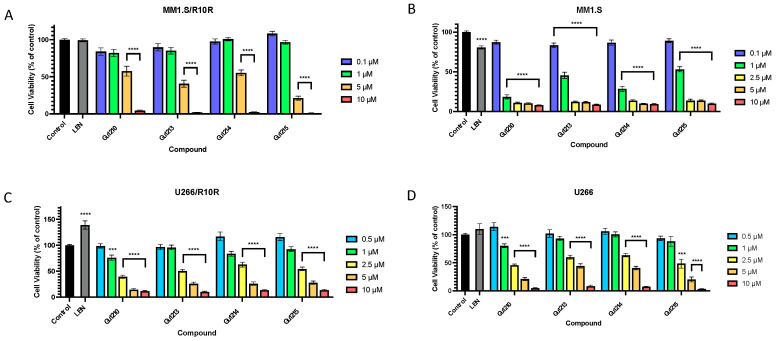
Treatment with Gu compounds inhibited multiple myeloma cell proliferation in vitro. (**A**) The 72-h treatment of MM1/R10R cells with the four lead analogs at a range of concentrations (0.1 μM–10 μM) showed a dose-dependent response, with Gu1215 exhibiting the most potent inhibition (**** *p* < 0.0001). (**B**) The 72-h treatment of MM1.S cells with the four lead analogs at a range of concentrations (0.1 μM–10 μM) showed a dose-dependent response, with Gu1210 exhibiting the most potent inhibition (**** *p* < 0.0001). (**C**) The 72-h treatment of U266/R10R cells with the four lead analogs at a range of concentrations (0.05 μM–10 μM) showed a dose-dependent response, with Gu1210 exhibiting the most potent inhibition (**** *p* < 0.0001; *** *p* < 0.001). (**D**) The 72-h treatment of U266 cells with the four lead analogs at a range of concentrations (0.05 μM–10 μM) showed a dose-dependent response, with Gu1215 exhibiting the most potent inhibition (**** *p* < 0.0001; *** *p* < 0.001). These data are representative of at least three independent experiments with at least three replicates per experiment. In all the figures, any cases in which the concentrations are grouped in a bracket indicate that all of the values within the bracket are individually statistically different from the vehicle control.

**Figure 4 biomolecules-14-00725-f004:**
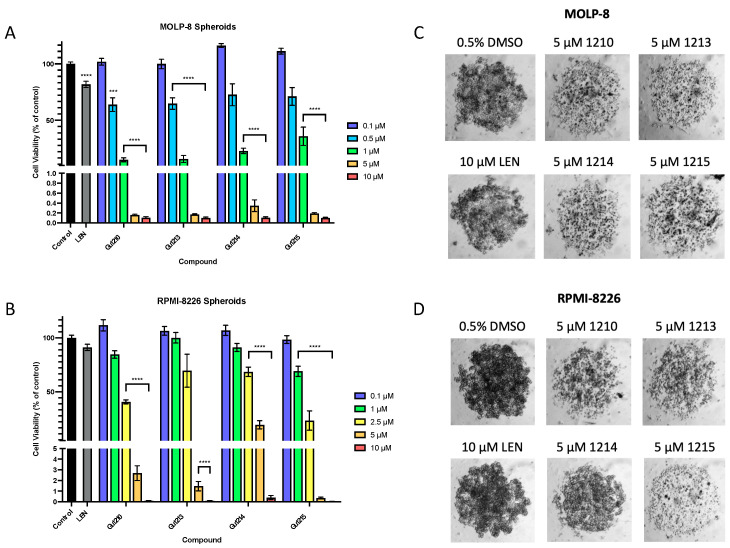
Treatment with Gu compounds reduced spheroid growth in vitro. (**A**) The 72-h treatment of MOLP-8 spheroids at a range of concentrations (0.1–10 µM) produced a dose-dependent response, with Gu1213 exhibiting the most potent inhibition of spheroid growth (**** *p* < 0.0001; *** *p* < 0.001). (**B**) The 72-h treatment of RPMI-8226 spheroids with the lead compounds at a range of concentrations (0.1–10 µM) also produced a dose-dependent response, with Gu1215 exhibiting the most potent inhibition of spheroid growth (**** *p* < 0.0001). (**C**,**D**) Representative images of the spheroids are shown. These data represent at least three independent experiments performed in triplicates. In all figures, any cases in which the concentrations are grouped in a bracket indicate that all of the values within the bracket are individually statistically different from the vehicle control.

**Figure 5 biomolecules-14-00725-f005:**
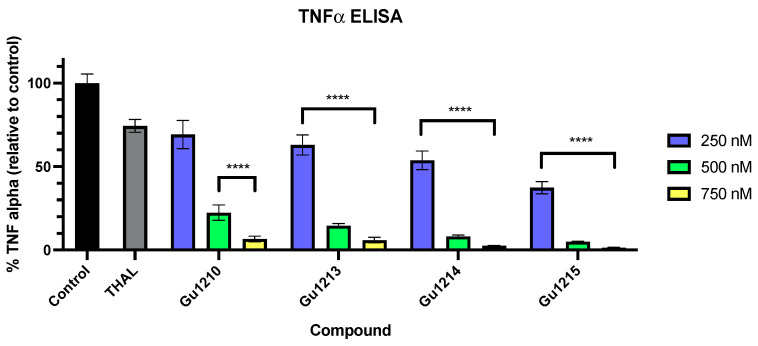
In vitro testing was conducted to assess Gu compounds’ effects on the inflammatory response in THP-1 cells, using a TNF-α ELISA. The vehicle control was 0.5% DMSO, while thalidomide (200 μM) was used as a comparator. THP-1 cells were exposed to the vehicle alone or with the compound for 1 h prior to the 3-h LPS challenge (1 ng/mL). Treatment of THP-1 cells with the four lead analogs (Gu1210, Gu1213, Gu1214, and Gu1215) at a range of concentrations (250 nM–750 nM) demonstrated a dose-dependent inhibition of TNF-α expression, with Gu1215 being the most potent inhibitor (**** *p* < 0.0001). The results shown are representative of at least three independent experiments with at least three replicates per experiment. In all the figures, any cases in which the concentrations are grouped in a bracket indicate that all of the values within the bracket are individually statistically different from the vehicle control.

**Figure 6 biomolecules-14-00725-f006:**
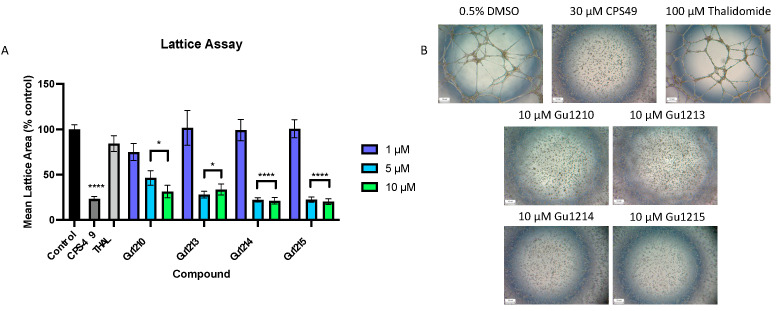
In vitro screening was conducted in an endothelial tube formation angiogenesis assay. The vehicle control was 0.5% DMSO, CPS49 (30 μM) was used as a positive control, and thalidomide (100 μM) was used as a comparator. (**A**) The graph represents the mean area of lattice formation relative to the vehicle control. The treatment of HUVEC cells with four lead analogs (Gu1210, Gu1213, Gu1214, and Gu1215) at a range of concentrations (1 µM–10 µM) showed a dose-dependent inhibition of tube formation (**** *p* < 0.0001, * *p* < 0.05). (**B**) Representative images of the tube formation assay. Results shown are representative of at least three independent experiments with at least three replicates per experiment. In all figures, any cases in which the concentrations are grouped in a bracket indicate that all of the values within the bracket are individually statistically different from the vehicle control.

**Figure 7 biomolecules-14-00725-f007:**
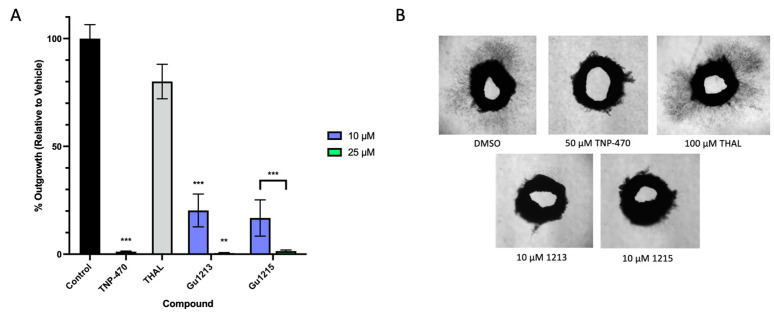
Ex vivo testing of thalidomide analogs in the human saphenous vein model of angiogenesis. The vehicle control was 0.5% DMSO, TNP-470 (50 µM) was used as a positive control, and thalidomide (100 µM) was included as a comparator. (**A**) The graph represents the mean area of microvessel outgrowth relative to the vehicle control. Fourteen-day incubation of human saphenous vein rings with Gu1213 and Gu1215 at 10 µM and 25 µM demonstrated a dose-dependent response, with Gu1215 exhibiting the most potent angiogenesis inhibition (*** *p* < 0.005, ** *p* < 0.01). (**B**) Representative images of human saphenous vein rings treated with the vehicle control, 50 µM TNP-470, 100 µM Thalidomide, or the indicated thalidomide analog at 10 M for 14 days. The results shown are representative of at least two independent experiments with at least two rings per experiment. In all figures, any cases in which the concentrations are grouped in a bracket indicate that all of the values within the bracket are individually statistically different from the vehicle control.

## Data Availability

The original contributions presented in the study are included in the article, further inquiries can be directed to the corresponding author.

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
