# Peer review of "Preclinical Evaluation of a Novel Series of Polyfluorinated Thalidomide Analogs in Drug-Resistant Multiple Myeloma"

_biomolecules, 2024, doi:10.3390/biom14060725_

Round 1
Reviewer 1 Report
Comments and Suggestions for Authors
biomolecules-3026091
The Manuscript " Preclinical evaluation of a novel series of polyfluorinated thalidomide analogs in drug resistant multiple myeloma" William D. Figg et al submitted to Biomolecules.
Authors have Previously published on the synthesis of polyfluorinated thalidomide analogs in ChemMedChem 2018, this is their further efforts to continuation of this work.
They have identified Gu1215 as a lead molecule out of four which was studied from their existing analogues. The study provides valuable insights into the development of novel therapeutics for multiple myeloma, addressing a critical unmet need in the field.
Authors have carefully drafted this manuscript with all supporting information and provided sufficient citation in the manuscript.
The manuscripts can be published, and it will be useful from scientific readers point of view.
Since lead compound Gu1215 where it has N-cyclohexane, does authors have any data with N-cyclopropane, N-cyclobutane, N-cyclopentane, N-cycloheptane?
Reviewer 2 Report
Comments and Suggestions for Authors
The paper describes the preclinical evaluation of new compounds based on thalidomide for the treatment of multiple myeloma.
I have several questions:
1. Please correct the very first sentence of the abstract. The name of the drugs is "immunomodulatory drugs".
2. Generally, in vitro and ex vivo are written in italics.
3. In cancer research, we talk about patients' survival, not longevity (line 32, p1]
4. The question is if new IMIDs are relevant with the onset of immunotherapy (monoclonal antibodies, CAR T cells) in the treatment of MM. The authors should address this. Further, pomalidomide is fairly useful. So, why do we need new IMIDs?
5. MM cell lines used in the paper (RPMI8226, MOLP8, MM1S, U 266) were derived from peripheral blood of patients and are thus not really a good model for MM. JJN 3 is a cell line derived from a patient with plasma cell leukemia which is a different disease.
6. The concentration of the compounds (10 uM) seems very high, even for a cell culture.
Comments on the Quality of English Language
See questions above.
Reviewer 3 Report
Comments and Suggestions for Authors
Nowadays drug resistance is very important topic in cancer. The authors’ choice to perform a study ‘Preclinical evaluation of a novel series of polyfluorinated thalidomide analogs in drug resistant multiple myeloma’, is novel and adds significant information to the literature.
Abstract section:
This abstract section is not well-written and adequately presented. Authors should clearly present the aim of the study, the rationale, methods, results and the knowledge gap driving conduction of this study.
Introduction section:
The authors’ efforts to summarize the definitions in this topic of interest is not sufficient. While the target disease is explained well, protease inhibitor drugs currently used in the clinic should be mentioned.
What provides the target inhibition or induction called Cereblon as a pharmacophore should be interpreted or at least information should be given by compiling it from the literature.
Material and Methods section: Authors should clearly address the following comments with regard to the material and methods section of the manuscript.
Information on whether thalidomide analogues are used for purification or characterization in the LC-MS method should be shared.
In characterization, device information and chromatogram data should be presented in the supplementary section for 1H NMR and 13C NMR characterization illumination. Otherwise, it seems difficult to use both novel and new concepts for these compounds. After proving the structures, the CAS scan is considered evidential. Information should be given about the conditions under which the compounds were analyzed and with which solvents.
Results
It was concluded that the base images of the actively occurring images and the images containing the preferred alkyl group and lipophilic substituent from the floristic images were more active, but they were not mentioned in this way. You should make revision on it.
Discussion section: Is not well presented and the authors’ efforts to summarize the evidence in this topic of interest is not sufficient. You shoud make revised again
The authors should compare their results with previous studies. What advantages does it provide? What could be the clinical effects of explaining these axis?
I hope the suggestions for the better of this prepared manuscript will be useful to you.
Round 2
Reviewer 2 Report
Comments and Suggestions for Authors
The authors answered my questions in the letter to the reviewer but did not make any changes in the manuscript. so, please do that.
Comments on the Quality of English Languageno comments
Author Response
Manuscript Submission ID: Biomolecules-3026091
Thank you for reviewing our manuscript entitled “Preclinical evaluation of a novel series of polyfluorinated thalidomide analogs in drug resistant multiple myeloma” by Barton et al.
We thank the Reviewers for their comments and changes have been modified as requested where appropriate in the newly revised manuscript. We have provided a point-by-point response to the reviewers’ comments.
Round 2 Comments:
Responses to comments from reviewer 2:
The authors answered my questions in the letter to the reviewer but did not make any changes in the manuscript. so, please do that.
Response: We have made the changes in the revised manuscript and the changes are highlighted in red in the manuscript with the page and line numbers listed below each response.
- Please correct the very first sentence of the abstract. The name of the drugs is “immunomodulatory drugs”.
Response: We thank the reviewer for the suggestion. The IMiDs is defined originally as immunomodulatory imide drugs and over time classified as a class of immunomodulatory drugs containing an imide group. In literature, both definitions have been used and we have referred to both names in our manuscript. >> This is done on page 1 (lines 13, 34).
- Generally, in vitro and ex vivo and written in italics
Response: We have followed the MDPI Style Guide Section 3.6 on Italics (https://www.mdpi.com/authors/layout#_bookmark15) and have made the recommended changes of not italicizing throughout the manuscript.
- In cancer research, we talk about patients’ survival, not longevity (line 32, p1].
Response: We thank the reviewer for the suggestion. We have made the recommended change on page 1 (line 32).
- The question is if new IMIDs are relevant with the onset of immunotherapy (monocloncal antibodies, CAR T cells) in the treatment of MM. The authors should address this first. Further, pomalidomide is fairly useful. So, why do we need new IMIDs?
Response: IMiDs are the cornerstone of treatment for patients with MM and used in therapeutic combinations at all stages of disease including approved uses as maintenance therapy after autologous stem cell transplantion. However, patients become resistant to ongoing therapy over time and inevitably relapse. The treatment landscape of MM is continuously evolving. Next-generation IMiDs including iberdomide and mezigdomide (which are structurally similar to currently FDA approved IMiDs) are in clinical development in IMiD-refractory disease. Even in the new era of immunotherapy with CAR T cells and monoclonal antibodies, the purpose of new treatment regimens in both newly diagnosed and relapsed setting is to achieve deeper and more durable responses. As patients with MM become exposed to the three main classes of therapy earlier in treatment and still experience relapsed/refractory disease, it is critical that we continue to add innovative treatment options to our arsenal that can potentially provide long-term disease control. Our lead compound is a novel IMiD that is cereblon-independent and can overcome IMiD resistance.
>> We have summarized our above response to the following sentences in the manuscript on page 1-2, lines 44-49): “However, most patients with MM relapse multiple times, with increasingly shorter remissions, and eventually develop disease refractory to all available therapies. To improve clinical outcomes for these patients with relapsed-refractory multiple myeloma (RRMM), novel treatment modalities are required to achieve deeper and more durable responses as well as provide long-term disease control.”
- MM cell lines used in the paper (RPMI8226, MOLP8, MM1S, U 266) were derived from peripheral blood of patients and are thus not really a good model for MM. JJN 3 is a cell like derived from a patient with plasma cell leukemia which is a different disease.
Response: We recognized that while no MM cell line is perfect and highly representative of MM patient tumors, cell lines remain the workhorse of MM research. Specifically, for thalidomide and IMiDs drug development efforts over the past two decades, the MM cell lines used in our paper were previously used to develop first and second generation IMiDs (thalidomide/lenalidomide/pomalidomide). We therefore used these MM preclinical models (with both intrinsic and acquired IMiD resistance) to investigate our novel IMiD therapeutics as a comparison to earlier IMiD analogs. The well-known and earliest-established lines RPMI-8226 and U-266 emerged at the top of the citation ranking for the frequency of use of MM cell lines reported in the literature. According to Cellosaurus (RRID:CVCL_2078), DepMap (DepMap ID: ACH-000653) and Wellcome Sanger Institute’s Cell Model Passport (SIDM01036), JJN3 is a plasma cell myeloma with sample site at the bone marrow (https://cellmodelpassports.sanger.ac.uk/passports/SIDM01036).
>> In our above response, we explain to the reviewer that most researchers who work in the field of multiple myeloma are aware that the above cell lines are commonly used in MM research. We have further clarified and described each MM cell line on page 2 (line 84), page 5 (lines 220-221), page 6 (lines 261-261) and page 7 (lines 271-274). We have also provided the RRID in the Methods section so that researchers can look up these cell lines in Cellosaurus.
- The concentration of the compounds (10 µM) seems very high, even for a cell culture.
Response: We thank the reviewer for the comment. We follow the standard NCI-60 cell line screening protocol which sets 10 µM as the initial single dose high concentration preclinical screen in cell viability assays.
>> This is a standard screening dose used to identify effective compounds.
We have summarized our above response to the following sentences in the manuscript on page 5, lines 221-224): “Prior to dose-response drug treatment experiments, we first conducted a preliminary screening of the 26 compounds at the single 10 mM concentration for their ability to inhibit the growth of the two intrinsically IMiD-resistant MM cell lines and to identify effective compounds.”
Round 3
Reviewer 2 Report
Comments and Suggestions for Authors
I have no further questions